# Dairy Cows’ Udder Pathogens and Occurrence of Virulence Factors in Staphylococci

**DOI:** 10.3390/ani12040470

**Published:** 2022-02-14

**Authors:** František Zigo, Zuzana Farkašová, Jana Výrostková, Ivana Regecová, Silvia Ondrašovičová, Mária Vargová, Naďa Sasáková, Ewa Pecka-Kielb, Šárka Bursová, David Sandor Kiss

**Affiliations:** 1Department of Nutrition and Animal Husbandry, University of Veterinary Medicine and Pharmacy, Komenského 73, 04181 Košice, Slovakia; zuzana.farkasova@uvlf.sk; 2Department of Food Hygiene, Technology and Safety, University of Veterinary Medicine and Pharmacy, Komenskéhgo 73, 04181 Košice, Slovakia; jana.vyrostkova@uvlf.sk (J.V.); ivana.regecova@uvlf.sk (I.R.); 3Department of Biology and Physiology, University of Veterinary Medicine and Pharmacy, Komenského 73, 04181 Košice, Slovakia; silvia.ondrasovicova@uvlf.sk; 4Department of the Environment, Veterinary Legislation and Economy, University of Veterinary Medicine and Pharmacy, Komenského 73, 04181 Košice, Slovakia; maria.vargova@uvlf.sk (M.V.); nada.sasakova@uvlf.sk (N.S.); 5Department of Animal Physiology and Biostructure, Wrocław University of Environmental and Life Sciences, Norwida 31, 50-375 Wroclaw, Poland; ewa.pecka@upwr.edu.pl; 6Department of Animal Origin Food and Gastronomic Sciences, University of Veterinary Sciences Brno, Palackého Tř. 1946/1, CZ-61242 Brno, Czech Republic; bursovas@vfu.cz; 7Department of Physiology and Biochemistry, University of Veterinary Medicine Budapest, István Utca 2, H-1078 Budapest, Hungary; kiss.david@univet.hu

**Keywords:** early lactation, mastitis, non-aureus staphylococci, biofilm, antibiotics, methicillin resistance gene

## Abstract

**Simple Summary:**

Dairy farmers and those involved in primary milk production face the challenge of maintaining the health of their animals. Although the level of technological support during milking has increased and appropriate veterinary measures are available, mastitis remains a major health problem in dairy cows as well as a major threat to dairy farm profitability. The results of this study indicate that, in addition to the major udder pathogens (*S. aureus*, *S. uberis* and *S. agalactiae*) causing dairy mastitis, non-aureus staphylococci (NAS) are a major risk to dairy cows during early lactation. NAS, such as *S. chromogenes*, *S. warneri* and *S. xylosus*, which were isolated from animals with clinical mastitis characterized by mild, moderate, or severe symptoms and persistent infections, had the highest representation of virulence factors (production of hemolysis, gelatinase and biofilm; the ability to hydrolyze DNA; resistance to antibiotics) in comparison to less virulent strains. In addition, isolates *S. aureus* and NAS showed resistance to one or more antimicrobials in 77.2%, and in 16 (11.7%) isolates, multi-drug resistance to three or more antimicrobial classes was confirmed. Due to the high resistance to β-lactam-antibiotics in two isolates of *S. aureus* and two species of NAS, the presence of a methicillin-resistant gene, *mecA*, was confirmed, which potentially poses serious complications for the treatment of dairy cows and a serious health risk to milk consumers.

**Abstract:**

This study investigated 960 Slovak and Czech spotted cattle from four different conventional (non-organic) dairy herds located in Eastern Slovakia and Czechia during early lactation (14–100 days after calving). Dairy cows were examined clinically; milk from fore-stripping of each udder quarter was subjected to sensory examination and assessed by the California mastitis test (CMT), and laboratory analyses of bacterial pathogens in milk, including virulence factors, were conducted. Positive CMT scores (1–3) for one or more quarters were detected in 271 (28.2%) of the examined animals. Out of 230 infected milk samples, representing 24.0% of all dairy cows, staphylococci (59.1% of positive findings) were the most commonly isolated organisms, followed by *E. coli* (11.3%), streptococci *Str. uberis* (9.1%) and *Str. agalactiae* (3.4%), and enterococci (6.1%). From 136 isolates of *S. aureus* (38 isolates) and non-aureus staphylococci (NAS; 98 isolates), virulence factors and their resistance to 14 antimicrobials were detected using the disk diffusion method, with PCR detection of the methicillin resistance gene, *mecA*. An increased incidence of clinical and chronic forms of mastitis has been reported in mastitic cows in which staphylococci, especially *S. aureus* and NAS (*S. chromogenes*, *S. warneri*, and *S. xylosus*), have been detected and compared to other isolated udder pathogens. From those species, *S. aureus* and isolates of NAS mentioned above showed multiple virulence factors that are more likely to hydrolyze DNA, hemolysis, produce gelatinase and biofilm, and have multi-drug resistance as compared to other less virulent staphylococci. Generally, the isolated staphylococci showed 77.2% resistance to one or more antimicrobials, in particular to aminoglycosides, β-lactams, macrolides, or cephalosporins. Isolates that showed the ability to form a biofilm were more resistant to more than one antimicrobial than isolates without biofilm production. Multi-drug resistance to three or more antimicrobial classes was recorded in 16 isolates (11.7%), and the presence of the *mecA* gene was also confirmed in two isolates of *S. aureus* and two species of NAS.

## 1. Introduction

Dairy farming and milk production are among the basic pillars of livestock production and comprise the second biggest agricultural sector in the EU, representing more than 12% of total agricultural outputs. In European countries, dairy farmers produce around 170 million tons of milk per year, 97% of which is cows’ milk and 3% milk from ewes, buffalo, and goats. The EU rules emphasize that raw milk must come from healthy animals because the safety of dairy products is important in decreasing the risk of foodborne diseases [1]. 

Despite efforts by farmers to improve cows’ breeding environments and hygiene levels during milking, inflammation of the mammary gland, known as mastitis, is still one of the main health problems for animals and it causes economic losses for breeders. Infectious and non-infectious influences can cause mastitis. Infectious influences are mostly of microbial origin, and up to 95% of mastitis is caused by pathogenic bacteria that penetrate the teat canal into the mammary gland. In comparison with most other animal diseases, mastitis differs in the fact that several diverse kinds of bacteria can cause the infection. These pathogens are capable of invading the udder, multiplying there, and producing harmful, inflammation-causing compounds [2,3]. 

The most common forms of mastitis are caused by agents from two groups of bacteria. The group of contagious pathogens (e.g., *Staphylococcus aureus*, *Streptococcus agalactiae*, and *Streptococcus dysgalactiae*) includes bacteria that survive and grow within the mammary gland (MG), and thus the infection can spread from infected to uninfected quarters and from cow to cow. This occurs most frequently during milking [4]. The group of environmental pathogens consists of a variety of species, including some staphylococcal species. They prosper in the environment, especially in the presence of cow feces and skin [5]. 

Of this group, *Str. uberis*, non-aureus staphylococci (NAS), and *E. coli* are the most important, with multiple strains of varying pathogenicity for animals and humans [6]. Cobirka et al. [3] reports that many mastitis pathogens, such as *S. aureus*, *Str. agalactiae*, and *Str. uberis*, can be classified as contagious or environmental pathogens because they can be transmitted through multiple routes—not only through contagious milk from infected cows or poor hygiene during milking, but also via bedding, urine, feces, and other contaminants.

In general, each mastitis case is believed to be caused by one primary pathogen, as only one bacterial species is identified in the milk samples from affected udders. However, it is not rare to detect simultaneous infections by two different pathogen species, and even three pathogens have been found in a small proportion of samples [7].

In recent years, *S. aureus* and NAS have been found to belong to the most common microorganisms causing mastitis in dairy cows [8,9,10]. The manifestations of the inflammatory process caused by staphylococci are different, as they depend on the degree of reaction of the udder tissue to injury or infection. The clinical or subclinical manifestations of mammary gland infection, as well as its further course, depend on the interplay of immunity with the immune response of the dairy cow and the concentration and virulence of the staphylococcal strains [8]. The signs of clinical mastitis (CM) are sudden onset with redness and swelling of the udder, with altered milk from the affected quarter, containing flakes or clots with higher somatic cell count (SCC), or having a watery consistency. On the other hand, subclinical mastitis (SM) is characterized by a lack of visible signs in the udder or the milk, but the infection is still present. Affected animals with SM are associated with reduced milk production in the range of 60–140 L per cow annually, with an increased SCC of over 200,000 cells/mL [3].

The main complications associated with the treatment of intramammary staphylococcal infection include the fact that many strains can exhibit multiple virulence factors, some of them related to the severity of mastitis, which is controlled by a complex network of transcriptional regulatory factors. The ability of staphylococci to form biofilms and resistance to antimicrobials is one of the key factors in virulence that facilitates the adherence and colonization of these pathogens in the mammary gland epithelium, with ineffective antimicrobial treatment [6].

The increase in resistance also occurs because, in addition to treating clinical cases of intramammary infection (IMI), the common routine on farms is to dry dairy cows across the board with antimicrobials. The largest proportion is administered for the intramammary treatment of CM and dry cow therapy. According to a study by Ferroni et al. [11], management practices are associated with increased antimicrobials used in dairy cows. The authors analyzed 101 beef and dairy cattle farms in central Italy and compared the overall average antimicrobial consumption over the course of one year. The total course of administered antimicrobials was three times higher in the case of dairy cows than in beef cows. The increased number was mainly related to the treatment of lactating and drying cows with antimicrobials.

The studies Holko et al. [12], Idriss et al. [13], and Zigo et al. [4], performed on Slovak dairy farms, confirmed the increased resistance of mainly udder pathogens (*S. aureus*, *S. uberis*, and *S. agalactiae*) as well as NAS to those antibiotics that are part of the intramammary applicators used to treat dry cows (see Appendix A). 

On the other hand, NAS are considered to be minor pathogens in dairy mastitis; however, there are studies published by several authors that emphasize their role in the development of MG inflammation [14,15,16,17,18]. This heterogeneous group of bacteria consists of 54 species, of which at least 42 have been isolated from bovine-associated habitats such as quarter milk, teat apices, and/or rectal feces from dairy cows. They are also abundantly present in the cow’s environment, with every habitat and niche having its specific NAS distribution. Research is still ongoing to unravel the species-specific ecology and epidemiology and to study the host–microorganism interaction [15]. 

An increase in the occurrence of NAS on farms was observed after a decrease in the incidence of mastitis caused by the main pathogens; the causative NAS showed increased resistance to common antibiotics and disinfectants. Compared to *S. aureus*, NAS usually exhibit a lower number of virulence factors. The essential factor of pathogenicity of NAS is biofilm formation, which allows them to survive the application of disinfectants and other sanitation procedures [18]. Nascimento et al. [14] reported that the NAS (*S. epidermidis*, *S. saprophyticus*, *S. hominis*, and *S. aerletae*) isolated from mastitic cows were resistant to the antimicrobials used to treat cows during lactation and were able to produce some staphylococcal enterotoxins. 

In particular, multi-resistant strains of staphylococci associated with resistance to more than one antimicrobial class are a serious risk to public health [9]. Recent studies also suggest that multi-resistant staphylococci, especially β-lactams antibiotics, indicate the presence of methicillin-resistant staphylococci (MRS), which have been identified in raw milk and dairy products, including cheeses [15,19]. According to the WHO, the opportunistic ability of MRS strains to cause mastitis is a threat to public health. They may become a source of zoonotic infections, serving as a potential source of antimicrobial resistance genes for humans in contact with dairy cows [16]; however, the opposite may also occur with humans being a source of MRS to cows [17]. Of the MRS of concern, *Staphylococcus aureus* (MRSA) is the species most widely reported; however, in several studies, NAS was also identified as MRS isolates [15,16,17,20]. 

In addition to the increased antibiotic resistance of staphylococci, Haveri et al. [20] and Vasil et al. [18] confirmed biofilm formation and hemolysis from infected milk samples and considered them as important virulence factors involved in the development of mastitis. Previous studies indicated the importance of staphylococci and their virulence factors in the pathogenesis of mastitis and its clinical presentation. Therefore, this study aimed at determination of the occurrence and etiology of mastitis in four dairy herds, with the detection of selected virulence factors such as the production of gelatinase, hemolysis, and biofilm, the ability to hydrolyze DNA, and resistance to antibiotics, with the detection of the methicillin resistance gene *mecA* in isolated staphylococci.

## 2. Materials and Methods

### 2.1. Monitored Dairy Farms 

The practical part of this study was carried out using four different dairy herds located on farms in Slovakia and Czechia. The four farms were selected because they practiced dairy cow breeding with conventional (non-organic) farming using nationals breeds of cattle, to facilitate the detection of contagious and environmental udder pathogens. Slovak and Czech spotted cattle are breeds of combined utility, used both for dairy and meat production, with milk yields of 6.5–7.5 kg × 10^3^ per lactation. Both breeds are part of the worldwide population of the Simmental breed, which is widespread in Slovakia and the Czech Republic. Two dairy herds from eastern Slovakia (Presov region) were investigated, ranging in size from 250 to 350 dairy cows of Slovak spotted cattle between the 1st and 4th lactation, with an average daily milk yield of 21.6 ± 2.4 L and 23.2 ± 3.1 L, respectively. In Czechia (Moravian-Silesian region) two dairy herds of Czech spotted cattle were investigated, ranging in size from 200 to 300 cows between the 1st and 4th lactation, with an average daily milk yield of 18.7 ± 2.8 L and 22.1 ± 3.9 L, respectively. 

The dairy cows investigated on all four farms were kept in a free housing system on straw litter, with ad libitum access to water. They were fed a mixed feed based on silage, hay, and concentrate, in agreement with the nutritional requirements of dairy cattle (NRC) [21]. The exact amount of feed was determined by lactation performance, and the rations met the nutritional requirements of cows weighing 650 kg and with average milk yield of 20–30 L per day. All cows were milked twice daily in parallel (BouMatic, Skjern, Denmark or herring bone (DeLaval, Cardiff, UK) parlors. 

### 2.2. Dairy Cow Selection and Udder Health Examination

A total of 270 cows from the first and 215 cows from the second Slovak farm, and 250 cows from the first and 225 cows from the second Czech dairy farm were monitored at the same time. The dairy cows were selected on the basis of the formation of production groups according to the stage of lactation (early lactation refers to 14–100 days of lactation) and the phase of nutrition, which were compiled by the zootechnician on each farm. The selected dairy cows of the same performance class (early lactation) were housed in individual husbandry groups, which included 45–90 animals on each farm. Each dairy cow from the selected husbandry group was given a comprehensive clinical examination, including a sensory examination and palpation of the udder. Additionally, milk from the fore-stripping of each udder quarter was subjected to sensory examination and assessed by the California mastitis test (CMT) (Indirect Diagnostic Test, Krause, Denmark) [22] with the collection of raw milk samples from positive cows [12]. Subsequently, from the 960 examined cows, 689 cows had a negative CMT score, and 271 cows, based on clinical manifestations and with a CMT score indicating trace or positive (score of 1–3), were chosen for aseptic collection of 12 mL mixed quarter milk samples for laboratory analyses of bacterial pathogens, according to Holko et al. [12]. The samples were cooled to 4 °C and immediately transported to the laboratory and analyzed on the following day.

Each mastitis case was assigned a corresponding mastitis grade according to the National Mastitis Council [23], with the mastitis grade categorized according to severity levels. Subclinical mastitis (SM) was detected by high somatic SCC using CMT evaluation without any visible abnormalities of the milk or apparent signs of local inflammation or systemic involvement. Clinical mastitis (CM) was classified as mild mastitis (CM1), being characterized by visible changes in secretion, moderate mastitis (CM2), showing local signs of inflammation of the mammary gland, and severe mastitis (CM3), also showing general signs such as fever, low temperature, loss of appetite, or inability to stand. Chronic mastitis or persistent mastitis was detected based on history (previous treatment) of clinical examination of the udder with a positive CMT score. 

### 2.3. Cultivation and Determination of Bacterial Pathogens 

The 10 µL aliquots of all milk samples were inoculated onto plates with esculin blood agar (Oxoid, Hampshire, UK) and MacConkey (MAC) agar (Oxoid, Hampshire, UK), and the plates were cultivated aerobically at 37 °C and checked after a 24-h and 48-h incubation period. If more than 2 phenotypically different colony types were present, the milk sample was considered contaminated and rejected. The primocultivated colony from blood agar and identification of *Staphylococcus* spp. were sub-cultured onto different selective bacteriological media (No. 110, Baird-Parker agar, Brilliance UTI Clarity Agar, Oxoid, Hampshire, UK) and incubated at 37 °C for 24 h. The esculin hydrolysis, pigment formation, catalase positivity (3% H_2_O_2_), Gram positivity, and creation of free or coupled coagulase were determined according to studies by Holko et al. [12] and Vasiľ et al. [18]. All presumptive *S. aureus* and NAS were identified by a matrix-assisted laser desorption/ionization (MALDI-TOF) biotyper (Bruker Daltonics, Leipzig, Germany). Mass spectrometry measurements were carried out on bacterial extracts prepared according to the manufacturer′s instructions. MALDI-TOF analysis started with spotting one colony onto a ground steel target (Bruker Daltonik GmbH, Leipzig, Germany), followed by air drying for 15 min. Each sample spot was then overlaid with 2 μL of matrix solution (saturated solution of α-cyano-4-hydroxy-cinnamic acid in 50% acetonitrile with 2.5% trifluoroacetic acid), and again subjected to air drying for 15 min. The identification of the relevant microorganisms consisted of importing the raw spectra obtained for individual isolates into the BioTyper software, version 2.0 (Bruker Daltonik GmbH, Leipzig, Germany). They were then analyzed without any further intervention by the user [24]. As a control for good quality, standards *S. aureus* CCM 4750 and *S. chromogenes* CCM 3386 (Czech Collection of Microorganisms, Brno, Czech Republic) were used.

The streptococci were determined on the basis of minute transparent colonies on blood agar and sub-cultured on Edward’s agar medium (Oxoid, Hampshire, UK). The pure colonies were described based on their growth, with color and classic morphological and hemolytic characteristics. Suspected streptococci microscopically appeared as Gram-positive cocci, either in long or short chains. Standard biochemical tests, including catalase, sodium hippurate, and esculin hydrolysis, were carried out according to El-Aziz et al. [25]. 

The presence of enterococci from primocultivation was confirmed by Gram-staining and sub-cultivation on MAC agar and SlaBa-plates agar (Slanetz & Bartley, Medium, Oxoid Ltd., Basingstoke, UK), with the growth and color of typical colonies. Confirmed colonies of *Streptococcus* spp. and important strains of family *Enterobacteriaceae* were biochemically identified at the species level using the STREPTOtest 24 and ENTEROtest 24 (Erba Lachema, Brno, Czech Republic) and evaluated according to the manufacturer’s instructions by the software TNW Pro 7.0 (Erba-Lachema, Brno, Czech Republic) with a probability of correct designations of the species above 90%.

### 2.4. DNase Test and Detection of Biofilm and Hemolysis in Staphylococci

Confirmed staphylococci based on MALDI-TOF analysis was exposed to deoxyribonuclease (DNase test) according to Hiko [26]. An overnight cultured colony was inoculated in the form of lines on DNase agar (OXOID Ltd., Basingstoke, Hants, UK) and incubated for 18–24 h at 37 °C. The culture was over-flushed with 1 mL 1 mole/mL hydrochloric acid. The strains were determined as DNase positive, based on a DNA digestion zone of clear transparence surrounding the culture (Figure 1). The formation of biofilm was determined by the phenotypic method by growth on Congo Red agar (CRA) according to Vasiľ et al. [27]. The production of slime by all strains was compared by cultivation of the staphylococcal strains on CRA plates consisting of 0.8 g of CRA and 36 g of saccharose in 1 L of brain-heart infusion agar after incubation at 37 °C for 24 h and subsequently overnight at room temperature. The slime-producing strains form black colonies on CRA, whereas the non-producing strains develop red colonies (Figure 1).

The ability of staphylococci to produce hemolysins was also determined. According to Moraveji et al. [28], types of hemolysis were phenotypically characterized based on the lysis zone of each staphylococcal isolate on plates of blood agar base supplemented with 5% sheep blood after 24 and 48 h incubation at 37 °C. 

### 2.5. Detection of Sensitivity to Antimicrobials in Staphylococci

The susceptibility of staphylococci (*n* = 136) isolated from milk from the investigated cows was tested in vitro against 14 antimicrobial agents. The susceptibility tests of isolates were carried out on Mueller Hinton agar using a standard disk diffusion procedure [29]. The antibiotic discs used in the current study were penicillin (PEN; 10 µg), ampicillin (AMP; 10 µg), amoxicillin (AMC; 10 µg), amoxicillin+clavulanic acid (AXC; 20/10 µg), ceftiofur (CEF; 5 µg), oxacillin (OXA; 1 µg), cefoxitin (CFX; 30 µg), ciprofloxacin (CPR; 5 µg), lincomycin (LNC; 15 µg), neomycin (NMC; 10 µg), novobiocin (NVB; 5 µg), rifaximin (RFX; 15 µg), streptomycin (STR; 10 µg), and tetracycline (TET; 30 µg). The zone of inhibition was recorded in millimeters, and results were interpreted as previously described. The determined diameters of the respective inhibition zones were evaluated (susceptible, intermediate, resistant) according to CLSI breakpoints [30]. In the tests, the tribes *S. aureus* CCM 4750 and *S. chromogenes* CCM 3386 (Czech Collection of Microorganisms, Brno, Czech Republic) were used as a control. The choice of antimicrobials reflects the range contained in a number of intramammary products to treat mastitis, which are available in Slovakia and Czechia.

#### Detection of the *mecA* Gene from Isolated Staphylococci

Phenotypical positive *S. aureus* (18 isolates from 38) and NAS (29 isolates from 98) based on their antimicrobial resistance to β-lactams antimicrobials were subjected to PCR to test for methicillin resistance. Total genomic DNA was isolated according to Hein et al. [31]. DNA quality was checked using a BioSpec spectrophotometer (Shimadzu, Kyoto, Japan). 

The source of DNA obtained as supernatant with centrifugation was used in PCR reactions using primers MecA1 (GGGATCATAGCGTCATTATTC) and MecA2 (AACGATTGTGACACGATAGCC) (Amplia s.r.o, Bratislava, Slovakia) for detection of the *mecA* gene according to Poulsen et al. [32]. Confirmation of the identity of the PCR products (527 bp) with the selected primers was in accordance with the instructions specified by GATC Biotech (AG, Cologne, Germany) using Sanger sequencing. The similarity of the DNA sequences obtained from the isolates with those available from the GenBank–EMBL (the European Molecular Biology Laboratory) database were determined using the BLAST program (NCBI soft-ware package). *S. aureus* CCM 4750 (Czech Collection of Microorganisms, Brno, Czech Republic) was used as a reference strain for PCR in this study.

### 2.6. Statistical Analysis

Data were entered into Microsoft Excel 2007^®^ (Microsoft Corp., Redmond, WA, USA) and analyzed using Excel, State 11, and SPSS version 20 (IBM Corp., Armonk, NY, USA). The dependence of the production of virulence factors on the most frequently isolated staphylococci from clinical, chronic, and subclinical mastitis was statistically analyzed using the chi-squared test with the significance level α = 0.05, critical value χ^2^ = 2.206, and testing value—G. Statistical independence between isolates with virulence factors and isolates without virulence factors within each species was confirmed when G ˃ χ^2^; the independence was not statistically significant when testing values of G < χ^2^.

## 3. Results

An examination of four dairy herds showed that, of 960 dairy cows examined during the early lactation phase (14–100 days of lactation), 689 cows (71.7%) had a negative CMT score and 271 cows (28.2%) had a CMT score of trace or 1–3 for one or more quarters. Of the mixed quarter milk samples taken from each examined cow based on the anamnesis, a CMT score of trace or 1–3 was identified as bacterial agents causing a clinical or subclinical mastitis in 230 (84.8%) samples, and 41 samples (15.1%) were identified as negative or contaminated. Based on the clinical examination of the MG, assessment of CMT, and laboratory diagnosis of milk samples, the prevalence of CM in the monitored first and second Czech dairy farms was 7.2% and 12.1%, respectively. In the monitored first and second Slovak dairy farms, the prevalence of CM was 5.9% and 10.1%, respectively. An increased incidence of chronic mastitis of 4.2% and 5.6% was reported in the second Slovak and Czech dairy farms, respectively, with a high prevalence of clinical forms of intramammary infection (Figure 2).

The results of the culture and identification of udder pathogens are shown in Table 1. Of the 960 samples taken from the four dairy farms, 230 were positive for udder bacterial pathogens. Of the positive sample, 136 cases (59.1% of the infected samples) contained the most commonly isolated staphylococci. The NAS represented the most commonly detected bacteria (42.6% of positive findings); *S. aureus* (16.5%) were the second most abundant pathogens, followed by *E. coli* (11.3%), streptococci (*Str. uberis*: 9.1%; *Str. agalactiae:* 3.4%), and enterococci (6.1%). The most common form of intramammary infection was subclinical mastitis, which accounted for 46.9% of all the positive samples. In addition, 37.8% of the positive samples were clinical cases of mastitis, classified as mild mastitis, characterized by visible changes in secretion with a high score of CMT (24.7% of all positive cases), moderate mastitis (10.5%), additionally showing local signs of inflammation of the mammary gland, and severe mastitis (2.6%), with general signs on the body and udder. NAS (16.5% of all CM), *S. aureus* (7.8% of all CM), and *Str. uberis* (3.8% of all CM) contributed the most to the occurrence of clinical mastitis. Based on previous anamnesis and actual examination of udder health, chronic mastitis (15.2% of all positive cases) was caused primarily by *S. aureus* (4.7%), NAS (3.5%), *Str. uberis* (3.0%), *Str. agalactiae* (1.3%), and mixed infections (1.7%). 

Table 2 summarizes, in descending frequency, the isolated strains of *Staphylococcus* spp., and indicates their role in the type of mastitis and the occurrence of selected virulence factors. *S. aureus* was isolated from 18 clinical, 11 chronic, and 9 subclinical cases of mastitis and thus appeared to be the causative agent most frequently isolated with the highest ability to report virulence factors in our study. Isolated strains of *S. aureus* from clinical, chronic, and subclinical forms showed hemolysis in blood plates, production of gelatinase, biofilm, and the ability to hydrolyze DNA. In two isolates of *S. aureus* from clinical mastitis, the *mecA* gene was detected. Eight species were isolated from NAS, with the following recorded as the most numerable species: *S. chromogenes* (22.4%), *S. warneri* (20.4%), *S. xylosus* (18.4%), *S. epidermidis* (9.1%), *S. haemolyticus* (7.1%), *S. hyicus* (10.2%), *S. capitis* (4.4%), and *S. piscifermentans* (4.4%). The representation of NAS in the individual forms of IMI was different. Most frequently, cases of subclinical mastitis (39.0%) were detected, caused predominantly by *S. xylosus*, *S. hyicus*, *S. warneri*, *S. hyicus*, and *S. epidermidis*. Clinical and chronic mastitis was detected in 37 cases (27.2%) and 8 cases (5.9%), respectively, caused in particular by *S. chromogenes*, *S. warneri*, and *S. xylosus*.

Of all CM and chronic mastitis with NAS, 30 cases involved the production of hemolysins, 9 the hydrolysis of DNA, 7 the production of gelatinase, and 18 involved biofilm production. The production of biofilm was also found in 9 isolates of NAS from subclinical cases of mastitis. The significance level of α = 0.05 was confirmed in the isolated stafylococci *S. aureus*, *S. chromogenes*, *S. warneri*, and *S. xylosus* from CM and chronic mastitis, which had the most numerous representations of virulence factors (production of hemolysins, gelatinase, the ability to hydrolyze DNA, and biofilm) in comparison to less virulent strains. In addition, the *mecA* gene was confirmed from one chronic case of mastitis in *S. chromogenes* and one CM case in *S. warneri*.

In 136 isolates of staphylococci, in vitro resistance to 14 antimicrobials was tested by the standard disk diffusion method (Table 3). Generally, low resistance was shown to tetracycline, amoxicillin reinforced with clavulanic acid, rifaximin, and cephalexin. In three and two isolates of *S. aureus*, intermediate sensitivity to tested aminoglycoside and β-lactams antimicrobials was observed, respectively. The intermediate sensitivity to aminoglycoside antimicrobials was observed in three isolates of *S. chromogenes*, two isolates of *S. warneri*, and one isolate of *S. xylosus*. In two isolates of *S. chromogenes* and one isolate of *S. warneri* and *S. sylosus*, intermediate sensitivity to β-lactams antimicrobials was observed. The obtained results presented in Table 3 show that the resistance of *S. aureus* and NAS, in particular to aminoglycosides and β-lactams, indicates the trend of increasing multi-antimicrobial-resistant pathogens in monitored dairy farms. High resistance to streptomycin, neomycin, ampicillin, penicillin, amoxicillin, and oxacillin was observed in tested strains of *S. aureus* and NAS. Tested *S. aureus* (5 isolates) and NAS (10 isolates) demonstrated resistance to oxacillin. Resistance in 37 isolates was observed to other β-lactam antimicrobials. 

Table 4 shows the phenotypic resistance profile of 136 isolates of *Staphylococcus* spp. from infected milk samples. Of the tested staphylococci, 105 isolates (77.2%) showed resistance to one or more antimicrobials. To one antimicrobial, 51 isolates (37.5%) were resistant. To two or more antimicrobials, 54 (39.7%) isolates of all tested staphylococci were resistant. Multi-drug resistance to three or more antimicrobial classes were recorded in 16 isolates (11.7%). Tested staphylococci showed multi-resistance to a combination of antimicrobial classes, such as aminoglycosides, β-lactams, macrolides, and cephalosporins.

In particular, staphylococci with biofilm-forming ability were resistant to the tested antimicrobials in most cases. Figure 3 is a comparison of resistance to more than one antimicrobial in isolates of staphylococci forming or non-forming biofilm at a time. Isolates *S. aureus* and NAS that showed the ability to form a biofilm were more resistant (66.7% in *S. aureus* and 51.8% in NAS) than isolates without biofilm production (29.4% in *S. aureus* and 25.7% in NAS). 

The 47 isolates (34.6% of all isolated staphylococci) in which phenotypic resistance was confirmed to β-lactam antimicrobials were tested by PCR for methicillin resistance with the detection of the *mecA* gene (Figure 4). The presence of *mecA* gene was confirmed in four isolates of staphylococci (two isolates of *S. aureus* and one isolate each of *S. chromogenes* and *S. warneri*), which at the same time showed resistance to both cefoxitin and oxacillin. Based on the results of our study, these isolates were considered as methicillin–resistant staphylococci (MRS). 

## 4. Discussion

Mastitis is currently one of the main health problems of dairy cows, despite the increasing advances in technology and veterinary measures. The incidence of mastitis is, of course, highly dependent on the lactation stage [7,33]. In our study, we monitored the prevalence and etiology of mastitis in four dairy farms during the early lactation phase. Cows in this lactation stage (14–100 days after calving) represent the largest group in farms because milk production depends on them. The dairy cow produces a quantity of milk representing 42–45% of the total milk produced during the first 100 days of lactation. With such an enormous milk production burden, cows are exposed to stress factors, such as hormonal changes associated with lactogenesis, reduced dry matter intake (which is in contrast to the desired increasing milk yield), increased lipomobilization of body reserves with a negative energy balance, and a change in body score [33].

All the above-mentioned risk factors affect the non-specific and specific immune system, in particular, the MG, through which pathogenic microorganisms penetrate more easily from the external environment. With the onset of intramammary infection, one of the indicators is an increased SCC [22], which was confirmed in our study. Of the 960 examined dairy cows, 689 (71.7%) cows, based on anamnesis, clinical examination, and evaluation of CMT, were negative, and 271 cows (28.2%) showed trace or positive CMT, with a score of 1–3. Cows with a high SCC were in 84.9% cases (230 positive cows from 271 examined cows) positive for the presence of an udder pathogen, which poses a significant risk to the health of the individual and the spread of infection to the environment. Generally, IMI begins when pathogens pass through the teat canal, interact with the mammary tissue cells, and multiply and disseminate in the cisterns and throughout the duct system. The manifestation of mastitis depends mainly on the degree of reaction of the udder tissue to injury or infection [34]. Of the 230 infected cows, 46.9% were diagnosed with subclinical mastitis, 37.8% clinical (mild (24.7%), moderate (10.5%) or severe (2.6%) signs) or chronic (15.3%) mastitis (Table 1). The prevalence of CM in the present study was 5.9% and 10.1%, and 7.2% to 12.1%, in the monitored first and second Czech and Slovak dairy farms, respectively (Figure 3). The prevalence of clinical mastitis in the monitored Czech and Slovak dairy farms was approximately at the same level, from which we can conclude that the farms apply generally accepted procedures related to the breeding and milking of Simmental cattle. 

According to Singha et al. [7], CM represents a serious health problem that can result in the reduction of milk yield, milk quality deterioration, treatment costs, involuntary culling, death, increased risk of antimicrobial resistance, and reduced animal welfare. Therefore, the prevalence of CM should be at the lowest level in lactating cows. Our results indicate that the prevalence of CM in monitored farms is in contrast with the studies of Silva et al. [35] and Rahman et al. [36], who reported the prevalence of clinical forms from 2.3% to 4.1% in lactating cows. 

Wentz et al. [37] reported that many cases of CM are caused by Gram-positive microorganisms (*Staphylococcus* spp. Or *Streptococcus* spp.), which was also confirmed in our study. However, bacteremia develops in a substantial proportion of cows with coliform mastitis, and about 20% of udder infections are caused by Gram-negative microorganisms, depending on the farm structure and hygiene status. This is consistent with our results, whereas SM and CM caused by *E. coli* accounted for 11.2% of infections from all infected cows. 

In a Finnish study focused on the detection and etiology of mastitis, Pyörälä and Taponen [38] point to a much higher risk of CM caused by *S. aureus* and NAS, which was also confirmed in all monitored dairy herds. Of the 230 infected samples, NAS (42.6% of positive findings) and *S. aureus* (16.5%) were the most frequently detected, in 136 cases (59.1%). The isolates of *S. aureus* and NAS accounted for 7.8% and 16.5% of CM; however, *S. aureus*, *S. chromogenes*, *S. warneri*, and *S. xylosus* often caused chronic mastitis due to persistent IMI. As the results of our study show, the incidence of clinical and chronic mastitis in the investigated Slovak and Czech farms caused by *S. aureus* and some NAS is higher compared to other udder pathogens (Table 2).

Consistent with the results of Persson Waller et al. [34], we assumed that the chronic IMIs caused by *S. aureus* (4.7%) and NAS (3.5%) in our study are predominantly persistent rather than new infections. It has been shown that many cows with *S. aureus* IMI in early lactation were already positive at drying off, which can precipitate a persistent, subclinical infection into CM in immunocompromised animals after calving [21].

High detected incidence of staphylococci in our results is consistent with the study by Holko et al. [12], who recorded a high incidence of NAS and *S. aureus* isolated from infected milk samples during the examination of 42 dairy farms in the west of Slovakia. The NAS represented 35.9% of positive findings and were the most commonly detected bacteria. On the other hand, the authors confirmed high resistance to aminoglycosides and β-lactam antimicrobials but without the presence of methicillin resistance genes, which contradicted our study.

In recent years, increasing studies have reported *S. haemolyticus*, *S*. *chromogenes*, *S. warneri*, and *S. xylosus* as the dominant strains of NAS isolated from mastitis in dairy cows [39,40,41]. In addition to subclinical forms of IMI, NAS has been largely isolated from CM [40], which was confirmed in our study. The CM mastitis with mild or moderate signs caused by NAS was associated with increased SCC, ability to form a biofilm, and resistance to aminoglycosides and β-lactam antimicrobials, especially to penicillin, amoxicillin, and oxacillin.

The increased incidence of staphylococcal infection in dairy cows also encourages the highest degree of pathogenicity in the production of more virulence factors, which are of crucial importance in persistent and CM cases [42,43]. These include, among others, cell wall-associated factors, different enzymes, and exotoxins that facilitate the infection pathway. For the individual virulence factors, co-production of hemolysins ß and δ in five species and single hemolysin δ in eight species of staphylococci were observed. Hydrolysis of DNase was detected in *S. aureus*, *S. chromogenes*, *S. warneri*, *S. xylosus*, and *S. haemolyticus* as well as the production of gelatinase. The staphylococci *S. aureus*, *S. chromogenes*, and *S. warneri* had the most numerous representations of virulence factors resulting in the increasing incidence of CM and persistent cases in comparison to strains with no virulence factors (Table 3).

The formation of biofilms is considered an important virulence factor of staphylococcal strains, as it facilitates their adhesion to biotic and abiotic surfaces [42]. The ability of these pathogens to form biofilms adhering to the MG epithelium helps them to evade immunological defenses and causes recurrent or persistent infections. From our results, the ability to form a biofilm was attributed mainly to *S. aureus* as well as seven species of NAS isolated from CM and chronic mastitis. In addition to *S. aureus*, the NAS that caused CM and chronic mastitis demonstrated the production of hemolysins, the ability to hydrolyze DNA, and resistance to antimicrobials as other important virulence factors. 

According to Perez et al. [43], the interaction between the production of hemolysins and biofilm can increase adherence of staphylococci to bovine mammary epithelial cells and their survival during the body’s immune response and antibiotic treatment. Our results confirmed the fact that the bacteria carrying this typical peculiarity are highly resilient to antimicrobials. In isolates of staphylococci with the ability of biofilm-producing confirmed by the phenotypic method on Congo red agar were more resistant to more than one antimicrobials compared to the isolates without biofilm formation at a time (Figure 4). 

Melchior et al. [44], in their study of staphylococci isolated from mastitis milk in cows, reported that biofilm production and resistance to antimicrobials were the most frequent virulence factors in strains isolated from CM. Increasing biofilm production was evident in strains from CM and repeat cases of mastitis after previous unsuccessful treatment. The IMI caused by *S. aureus* or NAS is difficult to treat, even with intramammary antimicrobials, so proper consideration should be given to the infections produced by biofilm-producing bacteria. 

The resistance to one or more antimicrobials in our study was detected in 105 isolates (77.2%) of staphylococci. Multi-resistant isolates for three or more groups of antimicrobial classes accounted for 11.7% (16 isolates). The tested staphylococci phenotypically showed multi-resistance to a combination of antimicrobial classes, such as aminoglycosides, β-lactams, macrolides, and cephalosporins (Table 4). In addition, the presence of β-lactam-resistant strains in our results indicates the presence of methicillin-resistant staphylococci (MRS) which was confirmed in 47 isolates (34.6%), and PCR was used to confirm the *mecA* gene in two isolates of *S. aureus* and one isolate each of *S. chromogenes* and *S. warneri*. All positive staphylococci (*n* = 4; 2.9%) with the *mecA* gene showed resistance to oxacillin and cefoxitin and were considered MRS. In a study by Khazandi et al. [45], when the whole genome was sequenced, they identified the presence of a *mecA* homologue in four oxacillin-resistant *S. sciuri* isolates. The homologue was not detected by conventional *mecA* PCR or cefoxitin susceptibility testing. However, in our study, MRS were also phenotypically confirmed, so we do not assume the presence of a false positive *mecA* homologue.

In the Czech study by Vyleťelova et al. [46], who investigated 1729 bulk milk and individual samples from dairy cows, ewes, and goats, the most frequent bacterial pathogens were *S. aureus* and NAS (*n* = 634; 36.7%). The species were also examined for antimicrobial susceptibility by using the disk diffusion method and the presence of the *mecA* gene by the PCR method. Among the resistant staphylococci, *S. aureus* (51%) was found to be the most frequent, followed by *S. epidermidis* (34.7%) and *S. chromogenes* (12.2%). The presence of the *mecA* gene was confirmed from 13 isolates of resistant staphylococci to β-lactam antimicrobials that occurred mainly in cow’s milk.

A similar study by Bogdanovičová et al. [47] monitored the prevalence and antimicrobial resistance of *S. aureus* from 50 dairy farms situated in the Czech Republic. From 261 raw milk and filtered milk samples, the authors detected positivity for *S. aureus* in 58 samples, 37 (14.2%) of which were isolated from raw milk and 21 (8.1%) from filtered milk. Resistance to β-lactam antimicrobials, especially to amoxicillin and oxacillin, was detected in the largest proportion (17.8%) of raw milk *S. aureus* isolates, followed by the isolates resistant to macrolides and tetracycline. Using the PCR method, methicillin-resistant *S. aureus* (MRSA) with the presence of the *mecA* gene was detected in four isolates obtained from raw milk samples and two isolates from filtered milk. Based on the previous two studies [46,47] and our findings, we can confirm that the occurrence of IMI caused by staphylococci, mainly *S. aureus* with increased resistance to β-lactam antimicrobials, is still a major problem in Czech and Slovak dairy farms. The occurrence of MRS with the presence of the *mecA* gene, which is in the range of 3–6% from isolates strains, is also worrying.

The WHO classified *S. aureus* as a high-priority pathogen, and it has gained the most attention among resistant staphylococci. However, methicillin resistance has also been described in several species of the NAS group [48,49]. In our study, we confirmed that two isolates of *S. aureus* demonstrated the *mecA* gene as well as one isolate each of *S. chromogenes* and *S. warneri*. According to Vinodkumar et al. [50], the NAS is thought to be a reservoir for numerous resistance genes, which could be transferred into the more pathogenic *S. aureus*. In particular, the presence of the antimicrobials and their metabolites in the environment may be the reason for the selection and spread of resistant isolates. This negative impact of the massive use of antimicrobials, especially in cows drying with a combination of an ineffective antibiotic (without antibiogram before application) and slow degradation in the udder, may potentially be a cause of increasing resistance and MRS in veterinary medicine.

MRS is resistant to almost all β-lactam antimicrobials, and infections caused by these pathogens result in unsuccessful or repeated treatment, increased SCC, and poorer outcomes. When it comes to MRSA as the cause of mastitis, Norway may be regarded as a naive country, as MRSA has been associated with bovine mastitis on only one occasion [16]. This contrasts with our results and the current situation in Belgium, where Bardiau et al. [51] found a similar occurrence of MRSA in 4.4% of milk samples from clinical cases of mastitis and Vanderhaeghen et al. [52] found MRSA in 9.3% of milk samples from farms experiencing *S. aureus* mastitis. Although our results showed a higher resistance of the tested staphylococci to β-lactam antimicrobials than in previous studies [16,51] we can state that the occurrence of MRS in the monitored farms was approximately at the same level.

Due to the increasing resistance of bacterial pathogens and the occurrence of MRS in veterinary medicine, the European Union plans to reduce livestock antimicrobial sales by 50% by 2030, based on the strategy “*Farm to Fork*” [53]. The urgently required or necessary administration of antimicrobials in veterinary medicine and the dairy sector remains possible in the future, but only if justified primarily on the results of targeted diagnostics, which, by means of anamnestic data, clinical examination, SCC from utility control, and culture of samples with antibiogram, reveal the health status of the udder and its physiological functions in each dairy cow. 

## 5. Conclusions

The present study clarifies that more than half of the IMIs were caused by staphylococci (59.1%), especially NAS (42.6%), followed by *S. aureus* (16.5%) in monitored Slovak and Czech dairy farms. In addition to *S. aureus*, *S. chromogenes*, *S. warneri*, and *S. xylosus* isolated from CM and chronic mastitis indicated a high degree of pathogenicity in the production of more virulence factors in comparison to other strains of NAS (*S. epidermidis*, *S. haemolyticus*, *S. hyicus*, *S. capitis*, and *S. piscifermentans*). Resistance to aminoglycoside and β-lactam antimicrobials was frequently detected in the tested staphylococci, possibly because these are the antimicrobials most frequently used in the drying and mastitis treatment of dairy cows. Isolates with the ability to form a biofilm were more resistant to more than one antimicrobial at a time compared to the isolates without biofilm production. Based on the phenotypic manifestation of antimicrobial resistance, detection of the presence of the *mecA* gene was confirmed in MRS (2.9%) in two isolates of *S. aureus* and two isolates of NAS (one isolate each of *S. chromogenes* and *S. warneri*).

We can state that *S. aureus* still accounts for the highest number of chronic or severe mastitis cases, as well as the number of virulence factors, but some NAS species could have the same aggressive potential based on their production of gelatinase, hemolysis, biofilm, hydrolyzed DNA, and multi-drug resistance. Knowledge regarding the virulence of both *S. aureus* and NAS species associated with bovine mastitis, especially in combination with resistance patterns and the presence of MRS isolates, is important for designing efficient prophylaxis and treatment guidelines to minimize the negative effects on milk yield and culling hazards in dairy cows.

## Figures and Tables

**Figure 1 animals-12-00470-f001:**
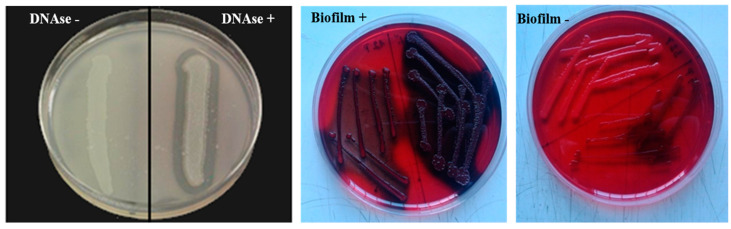
DNase test and biofilm production on Congo Red agar for staphylococcal detection. Source: Zigo et al. [6].

**Figure 2 animals-12-00470-f002:**
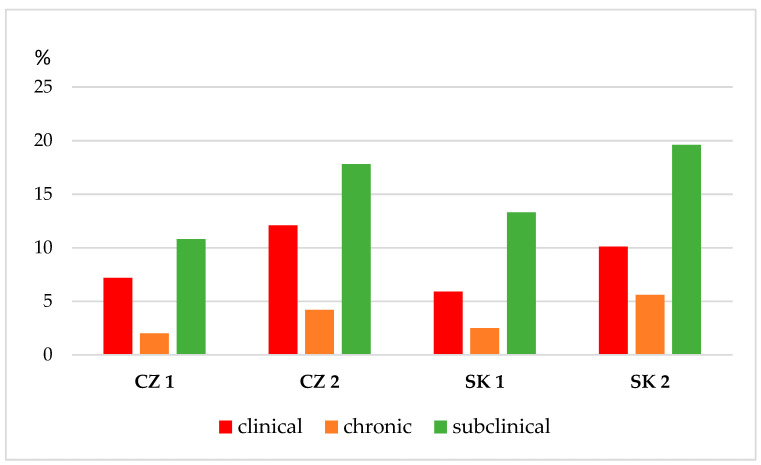
Prevalence of mastitis in monitored dairy herds during early lactation. Note: CZ 1–2: dairy farms situated in Czechia; SK 1–2: dairy farms situated in Slovakia. Subclinical mastitis: no signs are observed, the udder and milk appear normal, but an infection is still present with a positive CMT score and an increased SCC. Clinical mastitis: signs that are mild, moderate, or severe. Chronic mastitis: detected based on history (previous treatment) of clinical examination of the udder and positive CMT score.

**Figure 3 animals-12-00470-f003:**
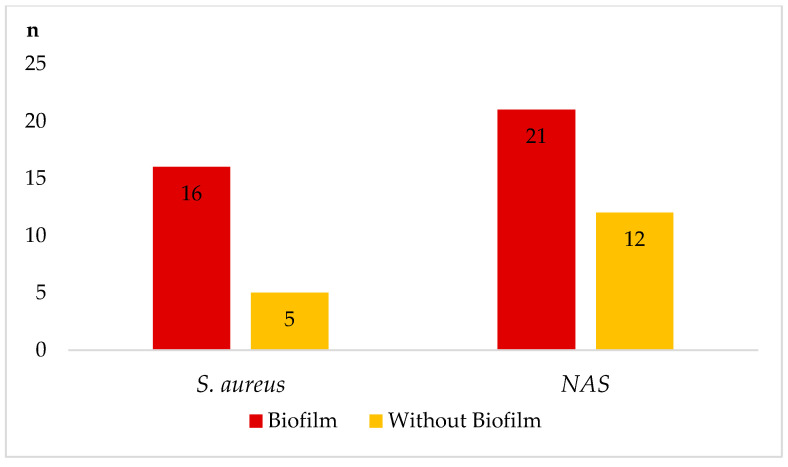
Comparison of resistance to more than one antimicrobial in isolates forming or non-forming biofilm at a time. Note: *n*—number of resistant isolates to more than one antimicrobial.

**Figure 4 animals-12-00470-f004:**
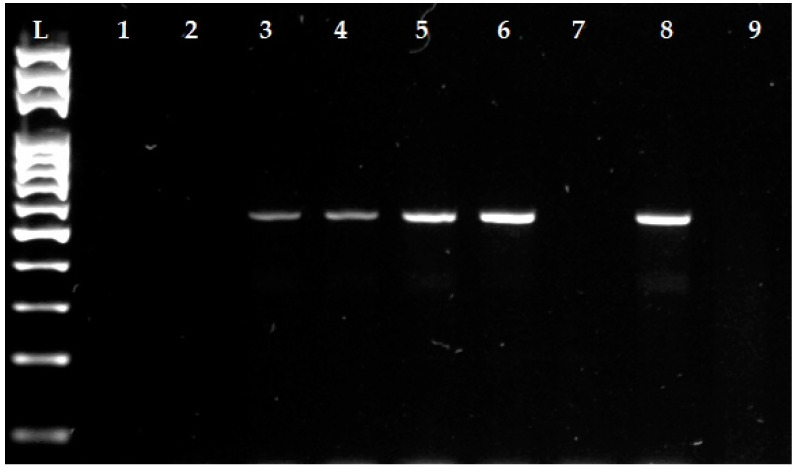
Detection of *mecA* gene in *Staphylococcus* spp. isolates isolated from infected raw milk samples using the PCR method (527 bp). L: 100 bp ladder; Line 1: isolate *S. xylosus* without *mecA* gene; Line 2: isolate *S. capitis* without *mecA* gene; Line 3: isolate *S. chromogenes* with *mecA* gene; Line 4: isolate S. *warneri* with *mecA* gene; Lines 5 and 6: isolate *S. aureus* with *mecA* gene; Line 7: isolate *S. haemolyticus* without *mecA* gene; Line 8: reference strain CCM 4750 *S. aureus* (positive control); Line 9: water (negative control).

**Table 1 animals-12-00470-t001:** Udder pathogens isolated from milk samples of four monitored dairy herds.

Pathogens	Number of Isolates	% (*n* = 230)	Clinical IMI ^1^ *n*/%			Chronic IMI ^2^	Subclinical IMI ^3^ *n*/%
			CM1	CM2	CM3	*n*/%	
NAS	98	42.6	28/12.2	7/3.0	2/1.3	8/3.5	53/23.0
*S. aureus*	38	16.5	10/4.3	5/2.2	3/1.3	11/4.7	9/3.9
*Escherichia coli*	26	11.2	5/2.2	2/0.9	0/0	2/0.9	17/7.4
*Streptococcus uberis*	21	9.1	4/1.7	4/1.7	1/0.4	7/3.0	5/2.2
*Streptococcus agalactiae*	8	3.4	0/0	3/1.3	0/0	3/1.3	2/0.9
*Streptococcus* spp.	10	4.3	4/1.7	0/0	0/0	0/0	6/2.6
*Enterococcus* spp.	14	6.1	2/0.9	1/0.4	0/0	0/0	11/4.8
Mixed infection ^4^	15	6.5	4/1.7	2/0.9	0/0	4/1.7	5/2.2
Total	230	100	57/24.7	24/10.5	6/2.6	35/15.2	108/46.9

Note: Clinical IMI ^1^—clinical intramammary infection (IMI), including mild (CM1), moderate (CM2) and severe forms (CM3) of mastitis; Chronic IMI ^2^—chronic or persistent intramammary infection; Subclinical IMI ^3^—subclinical intramammary infection. Mixed infection ^4^—include a mix infection of two bacterial pathogens.

**Table 2 animals-12-00470-t002:** The role of *S. aureus* and NAS in the form of mastitis and their virulence factors.

*Staphylococcus* spp./Number	IMI ^1^/Number	Hemolysins ^2^	DNAse ^3^	Gelatinase	Biofilm	*mecA* Gene	Testing Value
*S. aureus* (38)	clinical (18)	6α/4δ/1β	14	15	9	2	5.447 *
	chronic (11)	3α/2δ/2β	8	7	7	0	
	subclinical (9)	3α/1β	6	7	5	0	
*S. chromogenes* (22)	clinical (11)	4β/3δ	3	4	4	0	3.204 *
	chronic (4)	3β	1	1	2	1	
	subclinical (7)	2β/2δ	1	1	2	0	
*S. warneri* (20)	clinical (9)	4δ/2β	2	2	4	1	2.688 *
	chronic (3)	3β	0	0	1	0	
	subclinical (8)	3β/1δ	2	0	2	0	
*S. xylosus* (18)	clinical (7)	2δ/2β	2	0	3	0	2.255 *
	chronic (1)	0	0	0	0	0	
	subclinical (10)	4β/1δ	0	0	2	0	
*S. epidermidis* (9)	clinical (2)	1δ	0	0	1	0	1.012
	subclinical (7)	2δ	0	0	2	0	
*S. haemolyticus* (7)	clinical (4)	2β/1δ	1	0	2	0	0.742
	subclinical (3)	0	0	0	0	0	
*S. capitis* (6)	clinical (2)	2δ	0	0	0	0	0.401
	subclinical (4)	0	0	0	0	0	
*S. piscifermentans* (6)	clinical (2)	1β	0	0	1	0	0.851
	subclinical (4)	2δ	0	0	0	0	
*S. hyicus* (10)	clinical (0)	0	0	0	0	0	0.332
	subclinical (10)	1δ	0	0	1	0	

IMI ^1^—number of isolates and their influence on type of mastitis; hemolysines ^2^—production of hemolysin type α, β or δ; DNAse ^3^—ability of staphylococci to hydrolyze DNA; * Chi-squared test significance level α = 0.05; critical value χ^2^ = 2.206; Testing value (G) and statistical independence of virulence factors in isolated staphylococci was confirmed when G > χ^2^; the independence was not statistically significant when testing value was G < χ^2^.

**Table 3 animals-12-00470-t003:** Antimicrobial resistance of *S. aureus* and NAS isolated from mastitic milk samples.

Bacterial Strains (*n* = 136)	*S. aureus* (*n* = 38)		*S. chromogenes* (*n* = 22)		*S. warneri* (*n* = 20)		*S. xylosus* (*n* = 18)		Other NAS * (*n* = 39)	
Antibiotic/Resistance	%	*n*	%	*n*	%	*n*	%	*n*	%	*n*
Penicillin	18.4	7	13.6	3	20.0	4	11.1	2	10.3	4
Amoxicillin	18.4	7	18.1	4	15.0	3	11.1	2	10.3	4
Ampicillin	23.6	9	18.1	4	20.0	4	16.7	3	15.4	6
Amox. + clav.	0	0	0	0	0	0	0	0	0	0
Oxacillin	13.2	5	13.6	3	15.0	3	11.1	2	5.1	2
Cefoxitin	10.5	4	9.0	2	10.0	2	0	0	0	0
Cephalexin	10.5	4	9.0	2	0	0	0	0	0	0
Ciprofloxacin	10.5	4	9.0	2	10.0	2	11.1	2	0	0
Lincomycin	13.2	5	9.0	2	5.0	1	11.1	2	7.7	3
Neomycin	29.0	11	18.1	4	20.0	4	16.7	3	15.4	6
Novobiocin	18.4	7	18.1	4	10.0	2	77.8	14	7.7	3
Rifaximin	5.3	2	0	0	0	0	0	0	0	0
Streptomycin	29.0	11	27.3	6	25.0	5	11.1	2	17.9	7
Tetracycline	5.3	2	0	0	0	0	0	0	0	0

Note: Other NAS*: *S. epidermidis*, *S. haemolyticus*, *S. capitis*, *S. piscifermentans*, *S. hyicus*; *n*—number of tested isolates.

**Table 4 animals-12-00470-t004:** Phenotypic resistance profile in isolates of *Staphylococcus* spp. (*n* = 136) from mastitic cows.

Number Group of Antimicrobials	Phenotypic Resistance Profile	Number of Isolates	% of Isolates
0		30	22.1
1	STR	11	8.1
1	PEN	9	6.6
1	NMC	8	5.9
1	AMX	7	5.1
1	NVB	6	4.4
1	AMP	6	4.4
1	LNC	4	2.9
2	NMC, STR	9	6.6
2	AMP, NVB	4	2.9
2	CPR, NVB	2	2.2
2	LNC, NVB	2	1.5
3	PEN, AMX, OXA	4	2.9
3	AMP, AMX, OXA	3	2.2
3	PEN, LNC, NVB	2	1.5
3	AMP, OXA, NMC	3	2.2
3	AMP, AMX, NVB	4	2.9
3	PEN, AMX, AMC,	2	1.5
3 *	NVB, LNC, STR	4	2.9
4	AMP, CEP, FOX, OXA	3	2.2
4 *	RFX, CPR, STR, TET	2	1.5
4 *	CPR, LNC, NMC, NVB	3	2.2
4 *	NVB, CPR, NMC, STR	2	1.5
4 *	AMP, CEP, FOX, PEN	3	2.2
5 *	OXA, FOX, AMP, NMC, STR	2	1.5
Total ATBs resistant isolates		105	77.2

Note: * MDR—multi drug resistant isolates to three or more antimicrobial classes; AMX—amoxicillin, AMC—amoxicillin+clavulanat acid, AMP—ampicillin, CEP—cephalexin, CPR—ciprofloxacin, FOX—cefoxitin, LNC—lincomycin, NMC—neomycin, NVB—novobiocin, OXA—oxacillin, PEN—penicillin, RFX—rifaximin, STR—streptomycin, TET—tetracycline.

## Data Availability

All existed data are listed in the manuscript.

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
