# Peer review of "Dairy Cows’ Udder Pathogens and Occurrence of Virulence Factors in Staphylococci"

_animals, 2022, doi:10.3390/ani12040470_

Round 1

Reviewer 1 Report

This is a potentially interesting publication adding to the information on prevalence of mastitis pathogens and their AMR and virulence factor presence in Slovakia and Czech Republic. However, the purpose of the study is not clearly justified in the manuscript, and the publication is poorly written, therefore requiring major revisions. Additionally, the sample size is quite small, since only 4 farms were tested. What is the relevance of those farms and why were more not looked at?

The abstract and introduction are poorly written. There is a need for editing of the publication, looking at English grammar and meaning of the sentences. Since I am a reviewer and not a proofing editor, I will outline some of these issues and then move on to the content of the paper. Overall, this requires a major revision. There is less mistakes in materials and methods, but they should still be looked over and edited also. Results and discussion also contain mistakes, which includes potential calculation mistakes in the results.

The introduction seems to be a literature review or a book chapter, it lacks the typical wide to narrow format, setting the scene, and identifying a gap in research that this study is filling. Rather it describes much of the basics of mastitis, which should be beyond the scope of an original research paper introduction. It lacks the information I would like to see to understand why this paper is important – what is known about the prevalence of mastitis pathogens in Slovakia and Czech Republic, as well as the prevalence of clinical or subclinical mastitis. What are the rates of AMR in these countries and how they compare with the rest of Europe. And finally, how mastitis pathogens invade the mammary gland using the virulence factors that were studied here. This should lead to a clear formation of objectives. It should be stated why 4 farms were chosen, why specific animals in early lactation were chosen, and what the study is attempting to find out.

The final paragraph of the introduction needs to be improved and the rest of the introduction should be setting the scene for the final paragraph. This paper does not tell us more about how mastitis pathogens adhere to the host tissue and colonise the host, so sentence in lines 161-164 is unnecessary. The objectives need to be defined more clearly.  

Materials and methods lack certain details such as – when did the examination take place (what months, over what period of time), where were the farms located, and clarity on what stage of lactation were the cows in, why not all the cows were examined.

Presentation of results, including calculations in tables need to be revised or explained, more detail is included in line comments.

The discussion should include more comparisons to other studies of mastitis in Czech Republic and Slovakia, and  explain why only 4 farms and not all animals from those farms were used. There are also numerous prevalence and AMR studies on mastitis pathogens across Europe that this manuscript could be compared to. The discussion also often reviews the literature without conclusions or relations to this study, which is not the point of a discussion.

Please note that I did not correct all writing mistakes – please look through the publication carefully and correct all spelling and grammatical errors. I would also suggest trying to simplify the language as much as possible – in some cases formal words are used incorrectly, and also cutting down all the general paragraphs which do not contribute to the manuscript. This is a simple enough study, which should be a much shorter paper.

Line comments

line 23 – “what are resulted” --- this is grammatically incorrect, not sure what the meaning is supposed to be here but potentially change to “which resulted in”

line 35 – “were comprehensively investigated” --- either “960 cows were comprehensively investigated” or “we investigated 960 cows”

line 41 – “were most commonly isolated” --- change to either “the most commonly isolates organisms were” or “staphylococci were the most commonly isolated”. Please follow a pattern of placing a noun before a verb.

line 44 – “were detected virulence factors” --- change to “virulence factors were detected”. Please follow this logic and change these instances in the entire manuscript.

line 46 – italicise mecA

lines 47-48 – clinical mastitis does not demonstrate incidence of virulence factors – please rephrase the sentence, this is not A leads to B scenario, but rather A happens, and also B happens

line 49 – misspelled “haemolysins”

lines 50-51 – “Due to the high incidence of β-lactams-resistant staphylococci” – this should be rephrased – “due to” à “along with”. Even though it seems mecA was tested because of beta lactam resistance, this was not specified previously and does not belong to the abstract as it looks out of context. What was the “high resistance”, mention numbers. Also, it should be “beta-lactam-resistant” not “beta-lactams-resistant”.

lines 59-60 – “but EU rules emphasize that such products must come from healthy animals, which significantly limits their production and quality” --- Products coming from healthy animals limits their quality? It sounds like a complaint about the EU rules - generally, this sentence needs to be rephrased.

line 62 – “economic losses” – mastitis is not an economic loss -à “causes of economic losses”

line 66 – missing “the” in “the world’s dairy production”
line 67 – “that” is in twice

Please read through the publication and fix the grammatical, spelling and stylistic errors – I outlined some examples but following these lines I will not be pointing them out.

line 87 – Strep. uberis is both an environmental and a contagious pathogen

line 95 – “innate resistance and adaptive immunity” – I suggest to remove resistance and replace immunity with immune response – there is not enough knowledge on genetic resistance to mastitis, and authors do not mention genetic selection etc, plus environmental and farm management factors are at play here as well – so we are talking about an instance of immune response vs pathogen virulence, which determines the course of mastitis.

line 106 – ATB not explained in full text, only in the abstract. Also, not really necessary to abbreviate a word in this way – I would keep antibiotic.

lines 98-99 and then 108 – it is mentioned here that one of 2 clinical forms of IMI may develop in response to S. aureus, but then in line 108 a subclinical infection is described, not a clinical one. S. aureus can cause both clinical and subclinical disease and this should be mentioned. Also, peracute or subclinical mastitis are not the only 2 choices here, and such severe infections in response to S. aureus are very rare – this is a more typical E. coli type infection. Finally, a detailed description of peracute signs is not necessary, since this is an animal scientist audience and mastitis clinical signs are well-known.

Fig. 1 – how does this figure add to the story and is it based on data from one publication? If so, it should be cited in Fig. 1 caption. Also, it is not common practice to use someone else’s data to create a figure for publication – unless it collates information from various publications and is in a review article, rather than an original research article. I would remove this figure.

Table 1 – if category is DC in all cases, the column is redundant – please remove. Table 1 also provides an unnecessary level of detail for an original research article – and is never referred to again in the discussion, therefore there does not seem to be any relevance of this data to this paper. I would remove this table or include as supplementary table if authors deem it extremely necessary.

lines 113-132 – the mention of antimicrobial resistance and dry cow therapy could be shortened to 2-3 sentences. The authors do not look at dry cow therapy so there is little relevance of this to the paper.

lines 152-153 – “a source of zoonotic infections serving as a reservoir of antimicrobial resistance genes for dairy animals” – so is the issue a zoonotic infection or AMR in dairy animals? the main issue is that the MRS can be a source of AMR genes for humans in contact with dairy cows.

line 157 – “lysines” – lysine is an amino acid – is this what you mean?

lines 159 to 164 – Please add references to previous studies. Explain what is known about virulence factors and what they do. Please avoid general and unspecific phrases like “it is crucial to understand..” especially when it was not previously explained what the state of the art knowledge on adhesion, colonisation etc is – this paper did not identify what we know (or do not know), and therefore there is no point mentioning that we need to understand more. These sentences look rushed, and this is what most of the introduction should have been focused on.

lines 172-173 – Where were these farms located? Please also address how these farms are different from others previously studied.

line 174 – approximate size of 2 herds is mentioned, what about the other 2?

line 175 – “Fig1” – should be Fig. 2

lines 182-183 – How many cows were there per farm – did you investigate them all? If not, why not?

line 195 – first time subclinical mastitis is mentioned, should be covered in the introduction

lines 197-200 – It seems that clinical mastitis is further subdivided into 3 types, this needs to be mentioned more explicitly, right now the stylistic errors make this section difficult to read. Also, this subdivision is never mentioned in the results, so there seems to be little point in dividing clinical cases this way.

line 201 – what is “pcs”?

line 204 – what was the volume of milk collected

lines 230-233 – why were 2 cfus picked for S. aureus, S. agalactiae, but 5 cfus for others?

line 241 – please provide the full name of Strepto test and Entero test - MIKRO-LA-TEST ENTEROtest 24 N etc, and please specify whether it was the 16N or 24N kit. There are other Strepto tests available that do a completely different thing (identify group A streptococci only, for example). Also add “biochemically identified to the species level using…”.

lines 284-285 – what was the PCR product size? Did you check the quality of extracted DNA? It seems that the PCR product was sequenced – by what type of sequencing?

lines 300-301 – quarters had a CMT score of 1-3 – as in either negative, trace or positive (this scale was mentioned in methods)? and other quarters did not have a CMT score?

line 307 – you refer to figure 3, but caption (line 309) reads figure 5

line 331-332 and Table 2 – the numbers are slightly confusing. In Table 2, overall amount of clinical mastitis cases seems to amount to 53%, and so does the number of NAS in clinical mastitis cases in text. Same goes for subclinical mastitis – overall percentage of 230 cases is 47% and percentage of NAS that were subclinical is also 47%? It seems strange.

Because of this, it would be clearer if numbers as well as percentages were provided in Table 2 for species in clinical and subclinical mastitis. Also, the overall number and % of clinical and subclinical mastitis cases needs to be mentioned, as it is not included anywhere – and in Table 2 it would make more sense to calculate the percentages then with the overall clinical N or the subclinical N, instead of a percentage of overall cases.

line 336 – lysines – wrong word

line 337 – it’s “chi-squared”, not “chi-quadrate”

lines 337-339 – the explanation of the test is unclear. What is the hypothesis? What is “Test G”? G test is not the same as chi-squared. Please provide p values.

line 354 – it is unclear again what the null and what the alternative hypothesis was. The null hypothesis should be that there is no relationship between virulence factors and mastitis ie they are independent, since you stated in methods that you are testing for dependence. The sentence here: “The significance level of α = 0.05 was confirmed the independence of the production of virulence factors on severity of mastitis in cows” seems to suggest that the results were not significant, ie alpha was higher than 0.05, and null hypothesis was not rejected. If the null hypothesis is not rejected however, that does not mean it is true – it only means it could not be rejected, as in relationship could not be proven. You cannot say variables are independent based on that.

Tables 4 and 5 – What were the percentages calculated against in Table 4? For example, 18.4% of S. aureus is stated to be resistant to oxacillin. Based on the fact that there are 38 S. aureus isolates, that would be 7 oxacillin-resistant isolates. But in Table 5, the only OXA profile is in 4 isolates in total. Due to this being unclear please state number and % of resistant isolates in Table 4, or include a Supplementary Table with those numbers.

lines 379-381 – OXA and PNC not explained. Please include all these in alphabetical order.

line 382 – how many strains showed resistance to multiple beta-lactams and which beta-lactams?

line 385 – “presence was confirmed in 4 strains of staphylococci (2 strains of S. aureus, 2 strains of S. chromogenes and 1 strain of S. warneri)” – 2+2+1=5, not 4. Unless it is 4? Gel shows 4 + positive control.

Fig. 6 – was this the only gel/PCR performed? Ie, did you test 7 isolates only for the mecA gene? This should be mentioned in the results somewhere. Also, were the 4 (or 5, please clarify as per above) isolates containing mecA also resistant to oxacillin? Were there the only isolates resistant to oxacillin (as per inconsistency between Tables 4 and 5).  

lines 402-403 – “The incidence of clinical forms of mastitis in the present study was from 4,8% to 14,2% in Czech farms and from 6,5% to 14,7% in Slovak farms, respectively” – there are only 4 farms, so you are mentioning all prevalences on each separate farm, not a range. This sentence is misleading. It would be simpler to just present an overall prevalence – the same in results, since this exact same sentence occurs there.

lines 404-405 – But it was not stated anywhere in this paper that all these farms were tested in the same month, or that these clinical cases were notified anywhere – they could have occurred long before this study started seeing as mastitis can be chronic.

line 406 – stage, not period of lactation. Period of lactation is the entire lactation.

line 409-414 – so what? Were the cows examined in this study in early lactation? What do you conclude from this information? It is also important to note that early lactation is 14-100 days after calving (but also this depends on breed), and if this publication examines cows in the first 2 months following calving, then that is not all cows in early lactation, and could also include cows in post-calving (fresh cow) stage, which is when they are extremely vulnerable to new infections. Days post calving of cows tested should be mentioned in the study if the early lactation is of relevance to the results. Also in line 409 early lactation is incorrectly defined – it is not >100 days, it is <100 days.

lines 424-428 – The values mentioned here at the total S. aureus and NAS-caused cases from this study, not the clinical cases – which does not explain or prove “higher risk of clinical mastitis”. Also, based on Table 2 it seems that the number of clinical and subclinical cases caused by staphylococci are approximately the same, so that would disprove the fact that there is a higher risk of these pathogens causing clinical mastitis based on this study.

line 440 – “The NAS was mainly due to clinical mastitis” -  NAS cause clinical mastitis not are caused by it.

lines 453-466 – Explanation of biofilm and repetition of results is unnecessary as it does not lead to any conclusion.

lines 468-473 – This explanation is also unnecessary since it does not lead to any information related to this study. Was there a co-occurrence of haemolysins and biofilm formation? This study does not include or examine any information on the mechanisms of virulence and rather simply lists virulence factors, therefore the mechanistic explanation of how haemolysins and biofilm work together seems unnecessary.

lines 474-480 – Again, so what? How does this relate to this/your study? If no relation, paragraph is unnecessary.

lines 481-486 – but this study did not test for AMR genes, so it isn’t known whether this is resistance due to biofilm, or resistance due to AMR genes. Also, biofilm formation was tested here using Congo Red agar, which is an unreliable method. Limitations of this method should be mentioned.

lines 490-496 – unnecessary level of detail – how many multidrug isolates were found? Were they the same, or different antibiotics to those in this study?

line 500-502 – again inconsistency as to whether 5 or 4 strains with mecA were detected in total

lines 497-509 – This paragraph should be shortened.

lines 517-524 – Paragraph is unnecessary and not relevant to this paper as drying off was not studied here.

lines 525-538 – This is also unnecessary and does not relate to this study.

lines 540-541 – The fact that IMI is more frequent after calving and in early lactation is well-known – and this study does not compare to other stages of lactation so does not clarify that the majority of infections happen at the early lactation stage.

line 543- NAS did not have the highest “degree of pathogenicity”, S. aureus did

line 545 - “increasing impact of NAS on IMI” – this is something that should have been discussed previously, how is it increasing? CNS are common causes of IMI and can be the most prevalent pathogen type in other European countries.

“probably caused by reduced prevalence of other major causative agents” – what is this conclusion based on?

line 551 – “We can state that some NAS species have the same aggressive potential as S. aureus” – based on what?

Author Response

First of all, we would like to thank the reviewers for their helpful comments and valuable questions. Please find our comments and description of the changes below. All changes made to the manuscript have been highlighted in bubbles. As a result of these changes, the manuscript improved considerably – thank you!

General comments:

The manuscript was grammatically corrected by MDPI service. The abstract was supplemented by a number of multi-resistant isolates and a number of MRS. The introduction was shortened and figure 1 from previsions text has been deleted. Table 1 from introduction was added as supplementary material.

In material and methods were better described selected groups and lactation stage. Also, we explain the test hypothesis and define G value in statistical analysis. All tables and figures in results were rewritten and recalculated according to prevalence of clinical, chronic and subclinical mastitis forms. The resistance of tested staphylococci was described according to phenotypic characterization and antimicrobial classes – MDR. We add the comparison of resistance between biofilm-forming and non-forming staphylococci (fig. 3) In the discussion we shortened some paragraphs and corrected interpretation of obtained results. We also shortened the conclusion and corrected the interpretation of the findings.

Reviewer 2 Report

I sincerely hope my comments are taken as constructive criticism only.

It is my udenrstanding authors deliver realtively new information from a limuted number of farms in two neighbouring countries on the incidence of mastitis pathogens and their antimicrobial resistance profiles based on disk diffusion testing.

The manuscript will be interesting to read by many.  However, as currently prepared, the manuscipt requires extensive editing.  Additionally, few facts in the write up need significant changes.  Methodology used has nothing to do with the NMC guidelines and authors state that they followed these guidelines

Most of my comments are in the attached file.

Additionally 

  1. Authors have not mentioned the animal ethics committee approval at al.  This is essential for publication.
  2. The abstract must contain information on only 4 farms used and antimicrobial susceptibility based on disk diffusion testing
  3. Authors must consult native English speaker or MDPI editorial services to improve English quality.  Many sentences have lost the meaning with the current English quality.
  4. Authors do not adress any of the study limitations.  Addressing the study limitation do not decrease the value of the manuscript. It will only increase the value of it (e.g., small farm number, old technique used for testing of antimicrobial suscpetibility, presentation of multi-antimicrobial versus multi-drug resistant isolates, etc.)

Author Response

(The authors gave the same response as above.)

Round 2

Reviewer 1 Report

This manuscript has been improved tremendously. Grammatically, it is much easier to read. Methods, especially on selection of herds and cows are much clearer, and so is the introduction. Results seem to be expanded and are also clearer, with an effort put into the results tables. I would like to commend the authors on their effort, and have to say the merit and value of the publication is much clearer. Thank you for addressing my comments.

Small inconsistencies arised from the changes, which I point out in the attached file. The major aspect that I would like to see addressed is to not overestimate the virulence of NAS as opposed to S. aureus – while we can see a potential for increased virulence, based on the data from this publication the S. aureus strains were still more virulent – they caused more chronic and severe mastitis and had more virulence factors. Also, Table 4 has undergone a major change from the previous version – including for instance the number of isolates with no AMR being 77 previously, now reduced to 30. This is alarming, as it puts the integrity of the data into question. Could the authors explain why there is such a difference and perhaps share the raw data files behind the table?

Author Response

Dear reviewers,

thank you for fast re-evaluating our study and constructive comments. We have tried to incorporate all the recommended changes into the text.

Second round - Response to Reviewer 1 Comments

Abstract

line 49 – Increased incidence when compared to what?

Response: We add conclusion on base of table 1 when staphylococci were most causative agents of clinical and chronic mastitis.

line 52 – This is confusing - aren't S. aureus and NAS all of the staphylococci in the study? I would specify instead of saying NAS isolates - which/how many isolates were considered to be more virulent than which/how many. For example you could specify NAS species, as I am assuming that chromogenes, warneri and xylosus were considered to be the more virulent ones.
Or add "from those species", since it seems to me that this is a within species comparison.

Materials and methods

line 297 – I would specify here that these were chosen based on their antimicrobial resistance to multiple beta-lactams - or if I understand this wrong, please specify how they were chosen out of the 38 S. aureus and 98 NAS.

Response: We specified tested staphylococci by PCR based on their resistance to B-lactam antimicrobials.

lines 313-317 – This figure is a result so I would move down to results

Response: We renumbered and moved down figure 2 in part of results.

Results

line 343 – There is a Figure 2 above, all Figures below would need to be renumbered accordingly (unless Fig 2 is moved).

Response: We renumbered figures and corrected comments.

line 378 – Table 2: S. aureus still causes more % of clinical cases than chromogenes and warneri, and more chronic cases than warneri (with a comparable % to chromogenes). It also has more haemolysins, more DNASE, gelatinase and biofilm formers - this table does not show that NAS (even some of the species) are as virulent as S. aureus - which is later the conclusion.

Response: We corrected text in description of results obtained in table 2. The corrected text on base of reviewer comments is also in conclusion.

line 419 - To other β-lactam antimicrobials were demonstrated resistance in 37 isolates.

Response: We gramatically corrected text.

line  436 – Figure 3: Potentially would replace these % with absolute numbers, since percentages are mentioned in the sentence above.

line  450 – Table 4: Could you please explain why this table is so significantly changed, with new AMR profiles appearing?

Response: Table 4 was significantly changed after first round of revision because we cauculated only isolates which showed some virulence factor. Table 4 in present form contains data from all isolates with or without virulence factors based on results in Table 3.

Discussion

lines 470-475  – Repetition of results with no conclusions.

Response: We have rewritten the text and we added conclusion on the final paragraph.

lines 476-483 – Repetition of results with no conclusions.

Response: We have rewritten the text and we added conclusion on the final paragraph.

lines  498-503– Repetition of results instead of addressing whether the higher risk of CM due to S. aureus and NAS is also present here.

line  519 – references to the increasing studies needed

Response: We add references into text.

Lines 523-531 – Who is encouraged by the increased incidence and why?

Response: We corrected and rewritten text.

lines 557-666  – This paragraph repeats a lot of results with no discussion of them, perhaps it should be shortened or interspersed with the one below.

Response: We leave this paragraph because second reviewer would more discus about mecA homologue.

lines 568-584  – So, based on these, is the AMR increasing or decreasing in Slovakia/Czech Republic? This is a good review of other literature but what is the conclusion? Any differences between your studies?

lines 597-604  – And how does that compare to Czech republic and Slovakia, and what does your study bring to the table.

Conclusions

lines 616-620  – The highest among what - also how did you demonstrate that, there was no statistics between species only within species. I would rephrase to "a high degree of pathogenicity"

Response: We rephrased text.

lines 616-618  – But S. aureus still produced more virulence factors than NAS strains.

Response: We confirm in conclusion that S. aureus still produced more virulence factors than NAS strains.

line 625 – I would say "could have" instead of have, and "similar" rather than "same". As the paper is now written, it does not prove that NAS have the same aggressive potential as S. aureus - even if only some of the species are considered. S. aureus still comes on top in the number of chronic or severe mastitis cases, as well as the number of virulence factors or AMR.

Response: We corrected and rephrased text.

Reviewer 2 Report

Authors have addressed most of the comments from the previous version.  Few issues and corrections yet to be made (please see attached file)

Author Response

Dear reviewer,

thank you for fast re-evaluating our study and constructive comments. We have tried to incorporate all the recommended changes into the text.

Second round - Responses and comments to Reviewer 2

Simple summary

Line 33 – antibiotics

Response: After first roun of revision we tried to correct antibiotics to antimicrobials in whole text according to the reviewer's recommendation, but some parts escaped us.

Line 35– poses

Response: We corrected text.

Introduction

Line 66 – ewes, goats and buffalo

Line75 – channel

Response: We changed channel to canal.

Line 97 – In recent years,

Response: We changed references 9 for actual study.

Line 109 – Reference here required. A specific detail provided.

Response: We added reference.

Line 120 – of antimicrobials used in dairy cows

Response: We corrected text.

Line 143 – References here please.

Response: We added reference.

Line 156 – However, the opposite may also occur with humans being a source of MRS to cows (e.g. Grinberg A, Hittman A, Leyland M, Rogers L, Le Quesne B. Epidemiological and molecular evidence of a monophyletic infection with Staphylococcus aureus causing a purulent dermatitis in a dairy farmer and multiple cases of mastitis in his cows. Epidemiol Infect 2004; 132(3):507-13)

Material and methods

Line174 – ??? per lactation or per day

Response: We corrected milk yields per lactation.

Line 187 – Please stick to kg or L per day or per lactation. Changing in a single paragraph and throughout the manuscript is confusing. The worst part is kg cannot be directly converted into L and opposite.

Response: We leave milk production in liters (L) in whole text.

Line 202– (scores 1 - 3 based on XXX)

Response: Samples from positive CMT are defined in the following sentence – line 207.

Line 219 – Above you used "L'; here "l" for liter. Please be consistent throughout the manuscript.

Results

Line 367 – Table 1: Mixed infection

Response: We add description in note.

Line 378 – Table 2: Please explain in the caption.

Response: The Testing value is described and explained in a methods an a note.

Line 404 – Table 3: Enrofloxacin – not an antibiotic.

Response: For detection of antimicrobial resistance by disc difusion method we use Ciprofloxacin which is fluoroquinolone antimicrobial. In this case, enrofloxacin (is fluoroquinolone derivate) we used for injectabile treatment (Baytril, Bayer company) of clinical mastits cases with combination of intramammary drugs in monitored farms. We apologize for the substitution of antibiotimicrobials from this fluoroquinolone class.

Line 408 – no results in the table (3)

Response: We corrected text according to results in table 3.

Line 419 – To other β-lactam antimicrobials were demonstrated resistance in 37 isolates.

Response: We grammatically corrected text to…Resistance in 37 isolates was observed to other β-lactam antimicrobials.

Line 450 – Table 4: Number group of ATBs

Response: We corrected text according to the reviewer's recommendation.

Discussion

Line 509 – Reference here please

Response: We added reference.

Line 521– reduced milk production

Response: We corrected text according on the base of our findings.

Line 566 – The limitation of PCR to detect false-positive mecA analogue is not discussed (e.g. Khazandi M., Al-Farha AAB., Coombs GW, O’Dea M, Pang S., Trott DJ., Aviles RR., Hemmatzadeh F., Venter H., Ogunniyi AD., Hoare A., Abraham S, Petrovski K. Genomic characterization of coagulase-negative staphylococci including methicillin-resistant Staphylococcus sciuri causing bovine mastitis. Veterinary Microbiology. 219: 17-22, 2018 doi: 10.1016/j.vetmic.2018.04.004)

Line 576 – Clumsy sentence.

Response: We corrected text.

Line 595 – is a frequent

Response: We corrected text according to the reviewer's recommendation.
